# Benefits of jasmonate-dependent defenses against vertebrate herbivores in nature

**Ricardo AR Machado[1,2,3], Mark McClure[4], Maxime R Hervé[2,5], Ian T Baldwin[3], Matthias Erb[1,2]\***

[1]Root-Herbivore Interactions Group, Department of Molecular Ecology, Max Planck Institute for Chemical Ecology, Jena, Germany; [2]Institute of Plant Sciences, University of Bern, Bern, Switzerland; [3]Department of Molecular Ecology, Max-Planck Institute for Chemical Ecology, Jena, Germany; [4]School of the Environment, Washington State University, Washington, United States; [5]Institut de Génétique, Environment et Protection des Plantes, Le Rheu, France

**Abstract** Endogenous jasmonates are important regulators of plant defenses. If and how they enable plants to maintain their reproductive output when facing community-level herbivory under natural conditions, however, remains unknown. We demonstrate that jasmonate-deficient *Nicotiana attenuata* plants suffer more damage by arthropod and vertebrate herbivores than jasmonate-producing plants in nature. However, only damage by vertebrate herbivores translates into a significant reduction in flower production. Vertebrate stem peeling has the strongest negative impact on plant flower production. Stems are defended by jasmonate-dependent nicotine, and the native cottontail rabbit *Sylvilagus nuttallii* avoids jasmonate-producing *N. attenuata* shoots because of their high levels of nicotine. Thus, endogenous jasmonates enable plants to resist different types of herbivores in nature, and jasmonate-dependent defenses are important for plants to maintain their reproductive potential when facing vertebrate herbivory. Ecological and evolutionary models on plant defense signaling should aim at integrating arthropod and vertebrate herbivory at the community level.

**\*For correspondence:** matthias.
erb@ips.unibe.ch

## Introduction

In nature, plants are attacked by a multitude of herbivore species. By removing plant tissues and interrupting essential physiological processes, herbivory can negatively affect plant fitness, impose selection pressure, and thus drive the evolution of plant defenses (*Züst et al., 2012*; *Agrawal et al., 2012*; *Huber et al., 2016a*; *2016b*). Herbivory does not necessarily reduce plant survival and reproduction, however (*Bruelheide and Scheidel, 1999*; *Morris et al., 2007*; *Corbett et al., 2011*). For example, minor insect herbivory that occurs early in the growing season can induce plant defenses, and thereby ensure enhanced protection during times of greater herbivore abundance (*Agrawal, 1998*; *Kessler and Baldwin, 2004*; *Poelman et al., 2008*). Likewise, herbivory that reduces competitor abundance may benefit a given individual (*Agrawal et al., 2012*). Accordingly, measuring the impact of herbivory on plant reproductive fitness in an ecologically relevant setting (e.g., one that includes the full suite of native herbivores), is essential for identifying the selective forces underlying the evolution of plant defenses.

Most plant defenses are regulated by jasmonates (*Howe and Jander, 2008*). Interrupting the biosynthesis of these hormones results in decreased constitutive and induced production of volatile and non-volatile secondary metabolites, defensive proteins and structural defenses (*Thaler et al., 2002*;

**eLife digest** Plants are attacked by many different herbivores, including insects and mammals, and often produce toxins in response to protect themselves. Toxin production is regulated by plant hormones called jasmonates. It is commonly assumed this ability helps plants to survive and reproduce in nature. However, proof that a plant's own jasmonates (also known as "endogenous jasmonates") can increase a plant's fitness in the wild is lacking, especially in the context of attack by mammals.

Machado et al. have now asked whether endogenous jasmonates increase the fitness of coyote tobacco plants that were under attack by herbivores in their natural habitats in Southwestern Utah. Plants that lacked jasmonates were attacked more strongly by various herbivores, yet unexpectedly only the damage by mammals – including gophers, deer and rabbits – caused the plants to produce fewer flowers. Since plants with more flowers tend to produce more offspring, the number of flowers is a measure of a plant's fitness. Damage by insects, which are often seen as major enemies of plants, did not result in a significant impact on the number of flowers.

Laboratory experiments then revealed that damaging plants in a similar way to mammalian herbivores strongly reduced the plants' fitness. However mimicking insect damage did not have such a large effect. Finally, feeding experiments with cottontail rabbits revealed that jasmonate-producing plants are protected by higher levels of the nicotine toxin, which can explain why these plants fare better when attacked by mammals in nature.

Jasmonates are well known to regulate plant defenses and provide protection against a wide variety of herbivores. However, these new findings show that this only translates into fitness benefits for the plants against a subset of herbivores. A major challenge in the future will be to study how diverse communities of herbivores shape the evolution of plant defense signaling. Including larger herbivores, like mammals, into such experiments will be challenging but necessary to understand how plants survive in nature.

*Li et al., 2004*; *Paschold et al., 2007*; *Howe and Jander, 2008*; *Zhou et al., 2009*). Surprisingly, however, despite the availability of many different jasmonate deficient mutants (*McConn et al., 1997*; *Thaler et al., 2002*; *Zhou et al., 2009*; *Kallenbach et al., 2012*), fitness benefits of endogenous jasmonate production in herbivore-attacked plants in natural environments have not been demonstrated so far. Exogenous applications of jasmonic acid (JA) or methyl jasmonate (MeJA) are known to enhance plant defenses and fitness (*Baldwin, 1998*; *Thaler, 1999*; *Heil et al., 2001*), but these treatments remain difficult to interpret because they may not simulate dosages, localizations, and timings of endogenous jasmonate responses. Apart from inducing defenses, jasmonates can also reduce plant growth (*Baldwin, 1998*; *Hanley, 1998*; *Redman et al., 2001*; *Zavala and Baldwin, 2006*; *Bodenhausen and Reymond, 2007*; *Machado et al., 2013*), by interfering, for example, with the gibberellin and auxin signaling cascades (*Onkokesung et al., 2010*; *Yang et al., 2012*). This suggests that the evolution of jasmonate defense signaling is accompanied by growth suppression that may reduce fitness benefits under low herbivore pressure.

Our understanding of jasmonates in plant-herbivore interactions is also wanting because most research has focused on leaf-feeding arthropods (*Mafli et al., 2012*; *Falk et al., 2014*), and much less is known with respect to vertebrate herbivores, even though vertebrates are often the primary consumers in plant communities (*Paige and Whitham, 1987*; *Hodgson and Illius, 1996*). What we do know is that vertebrate herbivores can exert strong selective pressure on plants (*Collins et al., 1998*; *Becerra, 2015*) by influencing growth (*Paige and Whitham, 1987*; *Bergman, 2002*; *Liu et al., 2012*; *Ishihama et al., 2014*), structural defenses (*Abrahamson, 1975*; *White, 1988*; *Young and Okello, 1998*; *Takada et al., 2001*; *Wilson and Kerley, 2003*; *Young et al., 2003*; *Kato et al., 2008*), reproductive timing (*Zamora et al., 2001*) and mortality (*Veblen et al., 1989*; *Gill, 1992*; *Vila and Guibal, 2001*; *Saint-Andrieux et al., 2009*). We also know that vertebrates tend to avoid plants that are rich in secondary metabolites, including condensed tannins and phenolics (*Cooper and Owen-Smith, 1985*; *Owen Smith, 1993*; *Furstenburg and van Hoven, 1994*; *O'reilly-Wapstra et al., 2004*; *Jansen et al., 2007*; *DeGabriel et al., 2009*; *Rosenthal and Berenbaum,*

*2012*). Furthermore, it has been shown that silencing the production of the nervous toxin nicotine can engender increased leaf damage by vertebrate browsers in the field (*Steppuhn et al., 2008*). Despite these advances, it remains unclear to what extent endogenous jasmonates actually help plants to maintain their fitness when facing herbivore communities that include vertebrates.

To assess the importance of jasmonates in protecting plants from native arthropod and vertebrate herbivores, we studied three experimental *N. attenuata* populations in their native environment in the Great Basin Desert (United States). Each population consisted of a mix of jasmonate deficient and wild type plants. For each population, we characterized the damage that was caused by vertebrates and arthropods and then correlated damage patterns with plant flower production as a measure of the plant's reproductive potential. We then assessed the impact of simulated herbivory on jasmonate-dependent flower production and defoliation tolerance under glasshouse conditions. As part of this glasshouse work, we quantified primary and secondary metabolites in the specific plant parts that experienced herbivory and damage from the different kinds of herbivores in the field. Based on our findings we conducted a controlled feeding experiment to assess the impact of jasmonate deficiency on consumption rates by cottontail rabbits that resided in our study area. Finally, we conducted a complementation experiment with the same rabbits to assess whether nicotine was the main jasmonate-dependent deterrent affecting consumption rates.

## Results

### Jasmonate-deficiency increases arthropod and vertebrate damage and decreases flower production

To evaluate the impact of jasmonates on herbivory-dependent plant fitness, we established three experimental *N. attenuata* populations (henceforth called 'Lytle', 'Poplar' and 'Snow') across the field station of the Lytle Ranch Preserve (St. George, UT, USA; *Figure 1—figure supplement 1*). In each plot, at least 12 jasmonate-deficient inverted repeat allene-oxide cyclase (irAOC) plants and empty vector controls (EV, 'wild type') were planted. The irAOC line has been characterized previously (*Kallenbach et al., 2012*). Its herbivory-induced jasmonate levels are reduced by more than 95% (*Kallenbach et al., 2012*; *Machado et al., 2013*) while flower production is similar to WT plants in the absence of herbivore attack (*Machado et al., 2013*). Five to seven weeks after the establishment of the populations, we recorded herbivore damage and counted the number of flowers on each plant as a strong predictor of Darwinian fitness (*Van Dam and Baldwin, 2001*; *Glawe et al., 2003*; *Baldwin, 2003*). Across all three plots, we observed four main herbivore damage types: leaf removal, stem peeling, leaf chewing and leaf sucking/piercing. The different damage types are characteristic for different herbivores including deer, rabbits, wood rats, gophers, caterpillars, ants, mirid bugs and leafhoppers (*Meldau et al., 2009*; *Stitz et al., 2011*; *Kallenbach et al., 2012*; *Schuman et al., 2012*; *Đinh et al., 2013*; *Schäfer et al., 2013*). Across all plots, jasmonate deficiency significantly increased herbivore damage and decreased flower production (*Figure 1*). In the Lytle plot, jasmonate-deficient plants suffered more stem peeling and leaf removal by vertebrates, but similar arthropod damage compared to wild type plants. Jasmonate-deficient plants also produced fewer flowers. Likelihood ratio tests based on Generalized Linear Models (GLMs) showed that this effect was associated with the higher occurrence of vertebrate stem peeling in irAOC plants. In the Snow plot, irAOC plants suffered more vertebrate leaf removal and damage from leaf chewing and leaf sucking/piercing insects than the wild type plants, but only leaf removal by vertebrates was associated with a reduction in flower production. In the Poplar plot, no vertebrate damage was observed, and jasmonate-deficiency increased damage by leaf sucking/piercing arthropods, but did not decrease flower production (*Figure 1*, *Figure 1—figure supplement 2*). Overall, jasmonate deficiency increased vertebrate damage more strongly than arthropod damage, and only jasmonate-dependent changes in vertebrate damage translated into a decrease in flower production.

### Mimicking vertebrate damage reduces flower production more strongly than mimicking arthropod attack

To understand the impact of herbivore damage patterns and jasmonate-deficiency on plant flower production in more detail, we mimicked the different types of damage that we observed in the field in a controlled glasshouse experiment and quantified flower production over the entire flowering

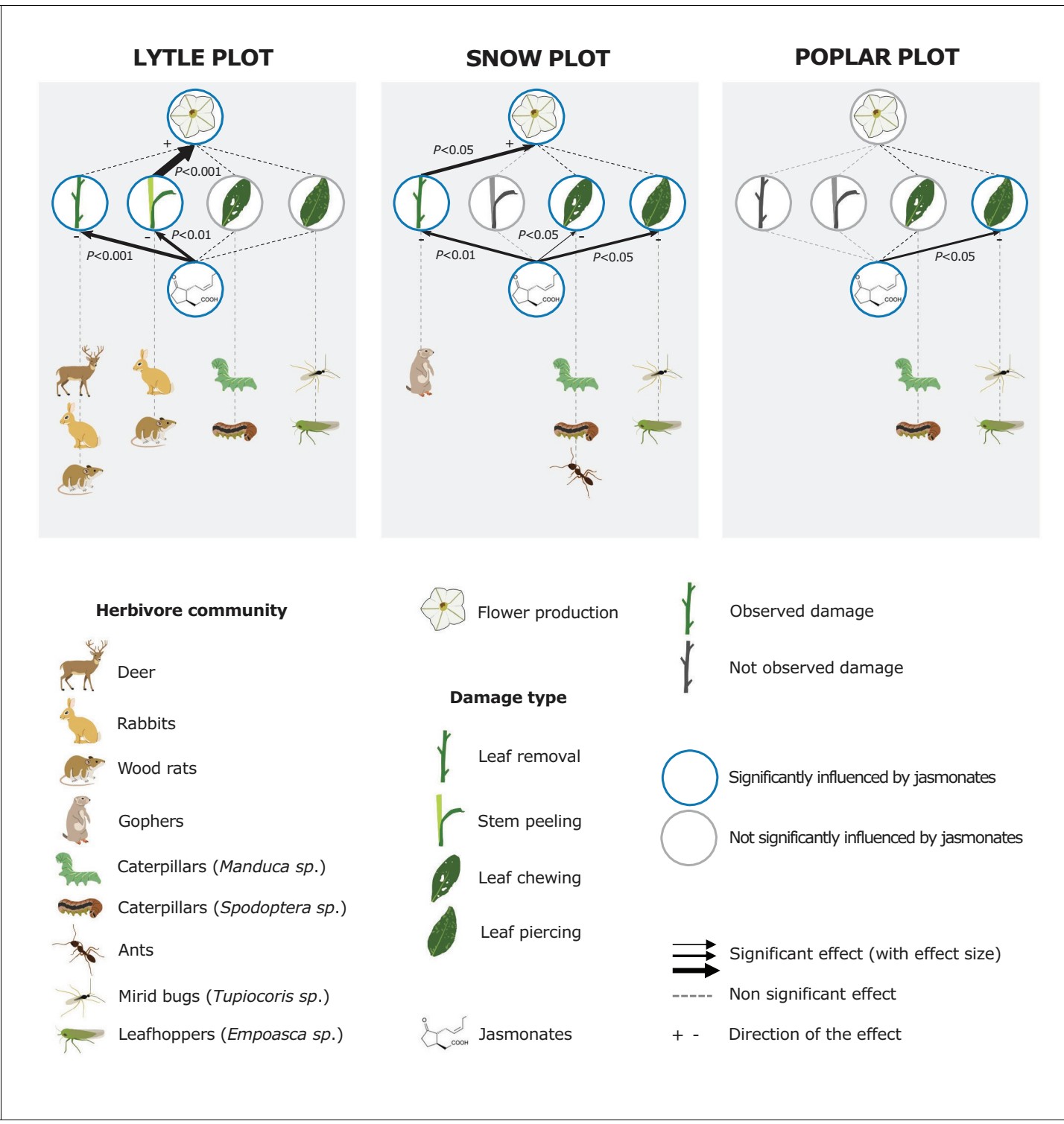

**Figure 1.** Jasmonate-deficiency reduces flower production by increasing vertebrate damage in nature. Effects of jasmonate deficiency on vertebrate and invertebrate damage by damage type and effect of damage type on flower production across three experimental plots ('Lytle', 'Poplar' and 'Snow') are shown (n = 12–19). Solid lines indicate significant effects of jasmonate-deficiency on herbivore damaage patterns and flower production. The herbivores responsible for the different damage types were identified based on field observations and characteristic feeding patterns. Jasmonate-deficiency increases damage by vertebrate and arthropod herbivores, but only vertebrate damage leads to a reduction of flower production as a strong predictor of plant reproductive potential.

*Figure 1 continued on next page*

*Figure 1 continued*

The following source data and figure supplements are available for figure 1:

**Source data 1.** Parameters used to determine the presence of different herbivores in the three experimental populations.
**Source data 2.** Quantification and herbivore association of the different damage types observed in the three experimental populations.
**Source data 3.** Leaf-herbivore damage screen and fitness measurements.
**Figure supplement 1.** Overview of the different experimental *N. attenuata* populations used in the present study.
**Figure supplement 2.** Detailed results of the analysis of the overall effect of jasmonate signaling and herbivore damage on *N. attenuata* flower production in the field.
**Figure supplement 3.** Photographic evidence of herbivore damage and the associated herbivores in the Lytle plot.
**Figure supplement 4.** Photographic evidence of herbivore damage and the associated herbivores in the Snow plot.

period (*Figure 2*). Attack by chewing invertebrates was mimicked by wounding (W) the plants with a pattern wheel and treating the wounds with *Manduca sexta* oral secretions (OS) (W+OS treatments) (*Machado et al., 2013*). Vertebrate damage was mimicked by removing either the rosette leaves (rosette defoliation), the rosette and stem leaves (full defoliation) or the rosette and stem leaves plus the stem bark (stem peeling). The artificially peeled stems looked similar to the vertebrate-damaged stems in the field, and histological staining revealed that stem peeling resulted in the removal of the epidermis and cortex from the stems (*Figure 2—figure supplement 1*). In wild type plants, W+OS induction did not significantly reduce peak flower production (*Figure 2a*). All other treatments led to significant reductions in flower production: rosette defoliation reduced flower production by 24%, rosette and stem leaf removal reduced flower production by 53%, and stem peeling resulted in a marked 78% reduction in flower numbers. Full defoliation and stem peeling also delayed the onset of flowering by more than two weeks. Overall, the same flowering patterns were observed in EV and irAOC plants (*Figure 2b*). However, the total number of flowers was significantly higher in defoliated and stem-peeled irAOC plants than in similarly treated EV plants.

## Jasmonates increase secondary metabolites and decrease carbohydrates in leaves and stems

As a first step to understand the mechanism behind the significant differences in vertebrate damage between EV and irAOC plants in the field, we profiled primary and secondary metabolites in the stems and leaves of glasshouse-grown plants. Redundancy analyses (RDAs) showed that simulated herbivory led to dramatic changes in both leaf- and stem carbohydrate profiles and secondary metabolite levels in a jasmonate-dependent manner (permutation test for the effect of the genotype x treatment interaction: $p < 0.001$ in both tissues) (*Figure 3* and *Figure 3—figure supplement 1*). The RDAs further revealed that diterpene glycosides (DTGs), rutin, nicotine, glucose and fructose in the leaves and nicotine, glucose and fructose in the stems explained most of the metabolic differences between induced EV and irAOC plants (*Figure 3*). Closer inspection of these metabolites showed that under control conditions, irAOC plants have lower nicotine and DTG concentrations in the leaves and lower nicotine levels in the stems than EV plants. Upon induction, these differences became even more pronounced (*Figure 3*). Constitutive rutin, glucose and fructose concentrations on the other hand did not differ between genotypes, but were more strongly depleted in induced EV plants (*Figure 3*).

## Jasmonate-dependent nicotine determines feeding preferences of cottontail rabbits

Feeding preferences by vertebrate herbivores are influenced by plant chemistry. Toxic secondary metabolites in particular can reduce food uptake (*Bryant et al., 1991*; *Iason, 2005*; *Rosenthal and Berenbaum, 2012*; *Camp et al., 2015*). To determine whether the nicotine deficiency observed in

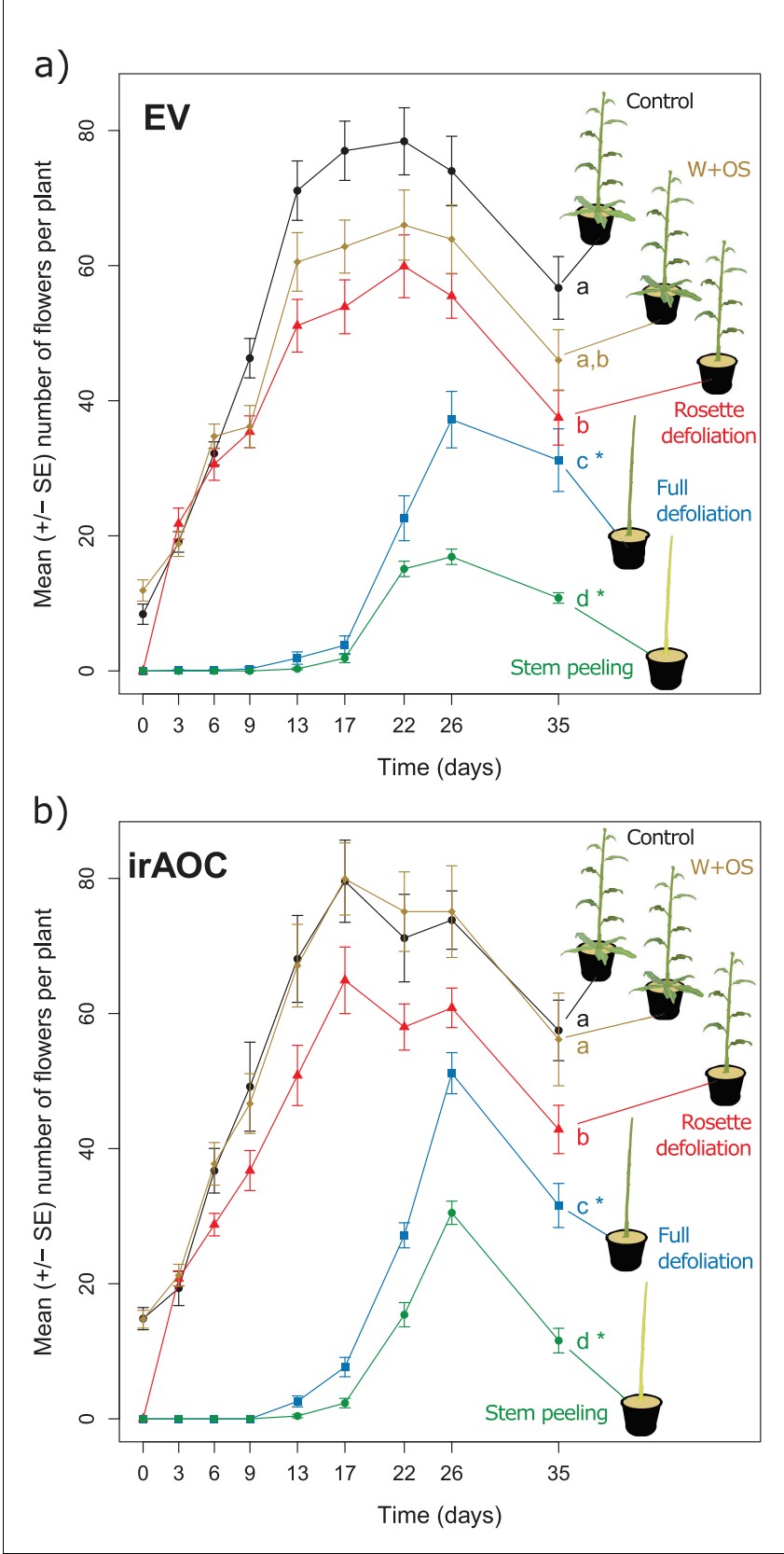

**Figure 2.** Simulated vertebrate attack strongly reduces plant flower production. Average (±SE) flower production of wild type (**a**) and jasmonate deficient irAOC plants (**b**) following different simulated herbivory treatments

*Figure 2 continued on next page*

*Figure 2 continued*
(n =10–12). Different letters indicate significant differences between treatments within genotypes (p<0.05).
Asterisks indicate significant differences between genotypes within treatments (*p<0.05). W+OS: Wounding +
application of *Manduca sexta* oral secretions.
The following source data and figure supplement are available for figure 2:
**Source data 1.** Fitness costs of induction and defoliation in the glasshouse.
**Figure supplement 1.** Toluene-blue staining of stem-peeled EV plants at the early bolting stage.

irAOC plants could explain the higher stem and leaf-removal by vertebrates in the field, we developed a biotechnology-driven *in vitro* feeding assay for cottontail rabbits (*Sylvilagus nuttallii*) (*Figure 4—figure supplement 1*). Cottontail rabbits can feed on *N. attenuata* in nature (*Karban, 1997*; *Baldwin, 2003*) and were present in our field experiment (*Figure 1—figure supplement 3*). In a first step, we produced food pellets by drying *N. attenuata* shoots of wild type and irAOC plants and pressing them into pellets. To determine whether drying affects *N. attenuata* defenses, we measured secondary metabolites in plants before and after drying (*Figure 4—figure supplement 2*). The drying process did not affect the water-loss corrected abundance of DTGs. The abundances of nicotine, chlorogenic acid and rutin were reduced, but jasmonate-dependent differences were conserved. The most abundant phenol amides dicaffeoylspermidine, dicoumarylspermidine, ferolylputrescine and caffeoylputrescine were degraded during the drying process, while less abundant phenol amides increased in concentration. From these data, we concluded that the dried plant material could be used to assess the impact of jasmonate-dependent alkaloids, DTGs and phenolic compounds, but not phenol amides, on vertebrate feeding preferences. In a first set of experiments, we presented eight individual cottontail rabbits with three types of food pellets: one consisting of pure dried alfalfa leaves as a rabbit maintenance food, and two others consisting of 1) a mixture of dried alfalfa leaves and EV or 2) dried alfalfa leaves and irAOC plant material at a ratio of 5:1 to account for typical food mixing by the rabbits. We then monitored food consumption as a response variable for feeding preference over 24 hr (*Figure 4*). Comparisons of adjusted confidence intervals of asymptotes of logistic models fitted for each treatment showed that cottontails rejected EV pellets and strongly preferred to feed from both irAOC pellets and pure alfalfa (*Figure 4a*, *Figure 4—figure supplement 3*). Similar feeding preferences were observed in two independent experiments (*Figure 4a*, *Figure 4—figure supplement 4a*). From our metabolite profiling, we had identified nicotine as a prominent candidate that might drive feeding preference of vertebrates. We therefore tested the hypothesis that nicotine may be responsible for the observed feeding patterns by complementing nicotine levels of irAOC pellets to match those observed in EV pellets (*Figure 4b*, inset). Strikingly, rabbits refused both nicotine complemented-irAOC and EV pellets and preferentially fed on alfalfa pellets (*Figure 4b*). Similar results were obtained in two independent experiments (*Figure 4b*, *Figure 4—figure supplement 4b*).

## Discussion

Since their discovery 25 years ago (*Johnson et al., 1990*), jasmonates have emerged as key signals that govern plant responses and resistance to herbivores (*Howe and Jander, 2008*). Early experiments demonstrated that the application of jasmonates strongly reduced herbivore damage in the field and increased seed production under high herbivore pressure (*Baldwin, 1998*; *Agrawal, 1999*). Experiments with jasmonate biosynthesis and perception mutants subsequently revealed that a reduction in endogenous jasmonate signaling increased plant susceptibility to a large number of organisms including detritivorous crustaceans (*Farmer and Dubugnon, 2009*), caterpillars of noctuid moths (*Bodenhausen and Reymond, 2007*), spider mites (*Li et al., 2002*), beetles (*Kessler et al., 2004*) slugs (*Falk et al., 2014*) and tortoises (*Mafli et al., 2012*). However, despite the increasing number of available jasmonate mutants in different plant species, it has remained unclear whether endogenous jasmonate production actually benefits plant fitness in nature. Filling this gap of knowledge is critical to understand the evolution and natural variation of defense

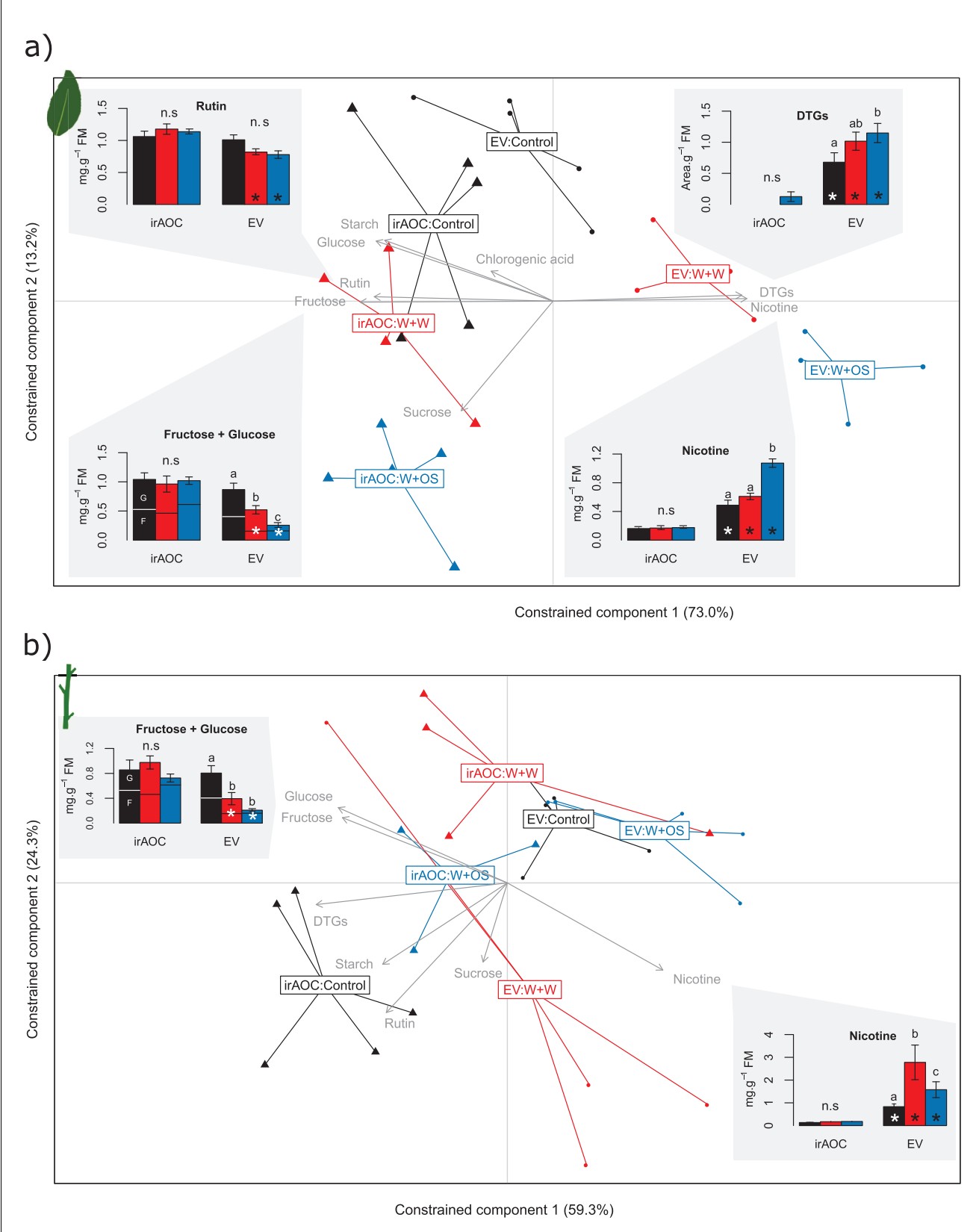

**Figure 3.** Herbivory-induced jasmonates increase secondary metabolites and decrease sugars in leaves and stems. Results of a redundancy analysis of the metabolic profiles in the leaves (**a**) and in the stems (**b**) of wild type (EV) and jasmonate-deficient (irAOC) plants are shown (n = 3–5). Insets depict

*Figure 3 continued on next page*

*Figure 3 continued*

average (±SE) concentrations of metabolites explaining most of the variation between induced EV plants and the other treatment*genotype combinations (correlation coefficients > |0.8|). Letters indicate significant differences between treatments within genotypes (p<0.05). Asterisks indicate significant differences between genotypes within treatments (p<0.05). Control: intact plants; W+W: wounded and water-treated plants; W+OS: wounded and *M. sexta* oral secretions-treated plants; DTGs: diterpene glycosides.

The following source data and figure supplement are available for figure 3:

**Source data 1.** Plant primary and secondary metabolite concentrations.

**Figure supplement 1.** Secondary metabolite profiles in leaves and stems of *N. attenuata* in response to simulated herbivore attack.

signaling (*Machado et al., 2013*; *Li et al., 2015*; *Xu et al., 2015b*; *Xu et al., 2015a*). Our experiments show that in the field, jasmonate-deficient plants were more strongly damaged by vertebrate and invertebrate herbivores. However, only in populations that experienced vertebrate herbivory did this effect translate into a reduction in flower production. Stem peeling in particular was associated with a jasmonate-dependent reduction in flower producion. Mimicking vertebrate and arthropod herbivory in the greenhouse confirmed that the type of damage caused by vertebrates has a strong negative impact on the plant's reproductive potential.

Plant damage at later developmental stages was not assessed in the present study. Furthermore, the outcrossing success of the different genotypes was not determined due to safety precautions related to the use of transgenic plants. Earlier studies demonstrate however that leaf damage at later developmental stages has little impact on plant fitness (*Zavala and Baldwin, 2006*) and that flower production in the absence of future damage is a valid and robust approximation for plant fitness in the predominantly self-pollinating *N. attenuata* (*Baldwin et al., 1997*; *Baldwin, 2003*; *Hettenhausen et al., 2012*; *Schuman et al., 2012*). Our results therefore highlight the importance of jasmonates in safeguarding the plant's reproductive potential under vertebrate attack.

Vertebrate herbivores can have a profound impact on plant fitness and the distribution of plant species in nature (*Hulme, 1994*; *1996*; *Shimoda et al., 1994*; *Palmisano and Fox, 1997*; *Waller and Alverson, 1997*; *Gómez and Zamora, 2000*; *Sessions and Kelly, 2001*; *Warner and Cushman, 2002*; *Maron and Crone, 2006*; *Maron and Kauffman, 2006*; *Nisi et al., 2015*). For instance, *Paliurus ramosissimus* trees that are intensively bark peeled by *sika deer* (*Cervus nippon*) will take nearly 30 years to recover (*Ishihama et al., 2014*). Similarly, strong growth delays and increased tree mortality have been observed in other species (*Veblen et al., 1989*; *Gill, 1992*; *Palmisano and Fox, 1997*; *Vila and Guibal, 2001*; *Saint-Andrieux et al., 2009*). Surprisingly however, vertebrate herbivores have rarely been considered in the context of plant defense signaling (*Dyer, 1980*; *Smith et al., 1991*; *Liu et al., 2012*). Instead, plant defense regulation has mostly been investigated in the context of arthropod herbivores and pathogens (*Dixon et al., 1994*; *Chen et al., 1995*; *Yang et al., 1997*; *Rojo et al., 2003*; *Wu and Baldwin, 2009*, *Wu and Baldwin, 2010*; *Schmelz, 2015*). Although arthropod damage in the field is generally low (*Steppuhn et al., 2004*; *Rayapuram and Baldwin, 2007*; *Pandey and Baldwin, 2008*; *Rayapuram et al., 2008*; *Johnson et al., 2009*; *Meldau et al., 2009*; *Stitz et al., 2011*; *Fischer et al., 2012*; *Kallenbach et al., 2012*; *Schuman et al., 2012*; *Schäfer et al., 2013*), local outbreaks can lead to substantial plant damage (*Van Bael et al., 1999*; *Cease et al., 2012*; *DeRose and Long, 2012*; *Đinh et al., 2013*), and several studies show that arthropods influence plant fitness (*Baldwin, 1998*; *Maron, 1998*; *Kessler and Baldwin, 2004*; *Machado et al., 2013*) and may drive the evolution of plant defenses (*Huber et al., 2016*; *Poelman and Kessler, 2016*; *Züst and Agrawal, 2016*) including plant defense signaling (*Durrant et al., 2015*; *Xu et al., 2015a*). Given the context-dependent potential of both arthropod and vertebrate herbivores to act as agents of natural selection, it seems crucial to include both groups when studying the ecology and evolution of plant defense regulation (*Strauss, 1991*; *Hulme, 1994*; *1996*; *Palmisano and Fox, 1997*; *Sessions and Kelly, 2001*; *Oduor et al., 2010*).

Numerous studies have found that plant secondary metabolites can influence vertebrate foraging patterns (*Freeland and Janzen, 1974*; *McArthur et al., 1995*; *Mithen et al., 1995*; *Dearing et al., 2005*; *Scogings et al., 2011*; *Mkhize et al., 2015*). Vertebrates tend to avoid plants that are rich in

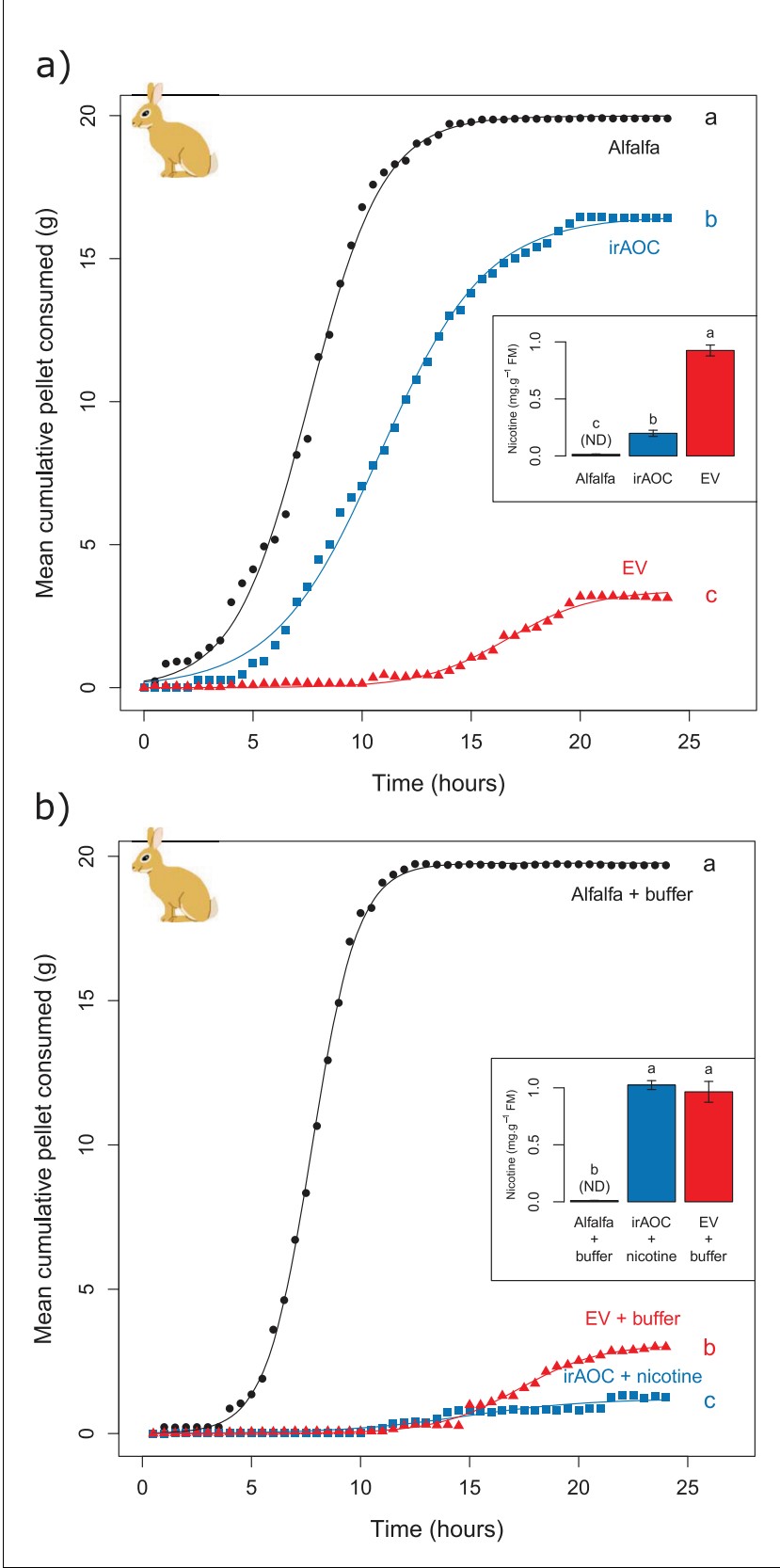

**Figure 4.** Jasmonate-dependent nicotine determines feeding preferences of a native vertebrate herbivore. Average amounts of food consumption by cottontail rabbits in a choice experiment. Choice experiments were *Figure 4 continued on next page*

*Figure 4 continued*
carried out with EV, irAOC and alfalfa pellets (a) (n = 8) or with EV, irAOC+nicotine and alfalfa pellets (b) (n = 9). Letters indicate significant differences in curve asymptotes tested based on asymptote confidence intervals of logistic models (p<0.001). Insets: nicotine contents of the different pellets. Letters indicate significant differences (p<0.05). N.D: not detected.

The following source data and figure supplements are available for figure 4:

**Source data 1.** Impact of drying on secondary metabolite profiles.
**Source data 2.** Nicotine concentrations in pellets.
**Source data 3.** Results of choice experiment (repetition).
**Source data 4.** Results of nicotine complementation experiment (repetition).
**Figure supplement 1.** Experimental setup for feeding preference assays for cottontail rabbits (*Sylvilagus nuttallii*).
**Figure supplement 2.** Secondary metabolite profiles in the leaves of *N. attenuata* before and after drying.
**Figure supplement 3.** Asymptotes and confidence intervals of the feeding preference assays for cottontail rabbits (*Sylvilagus nuttallii*).
**Figure supplement 4.** Results of the repetition of the feeding preference assays for cottontail rabbits (*Sylvilagus nuttallii*).

condensed tannins, phenolics and alkaloids (*Cooper and Owen-Smith, 1985*; *Owen Smith, 1993*; *Furstenburg and van Hoven, 1994*; *O'reilly-Wapstra et al., 2004*; *Jansen et al., 2007*; *Steppuhn et al., 2008*; *DeGabriel et al., 2009*; *Rosenthal and Berenbaum, 2012*): Quinolizidine alkaloids for instance seem to protect lupin (*Lupinus* spp) plants from rabbits (*Oryctolagus* spp.) (*Wink, 1985*; *1988*; *Fattorusso and Taglialatela-Scafati, 2008*). However, apart from toxic secondary metabolites, primary metabolites such as carbohydrates can also affect mammalian feeding preferences (*Mayland et al., 2000*). In this study, leaves and stems of jasmonate-deficient plants consistently contained lower levels of nicotine and higher levels of glucose and fructose compared to wild type plants. While both processes may have influenced vertebrate feeding preferences, we hypothesized that nicotine as a strong nervous toxin may have played a dominant role in increasing vertebrate feeding in jasmonate-deficient plants. The experiments with captive cottontails support this inference: Cottontails refused to feed on pellets containing 20% wild type *N. attenuata* shoots, but readily fed on pellets containing jasmonate-deficient, and thus nicotine-deficient, irAOC shoots. When irAOC pellets were complemented with nicotine to match the concentration in wild-type plants, rabbits avoided the two pellet types and only consumed alfalfa. This finding suggests that the amount of nicotine produced via jasmonate signaling is sufficient to dictate cottontail feeding preferences. From the plant's perspective, these results illustrate that nicotine effectively protects *N. attenuata* against sympatric vertebrates in a similar manner as it does against leaf feeding arthropods (*Steppuhn and Baldwin, 2007*). They further suggest that the increased damage and reduced flower production in jasmonate-deficient plants in the field can be attributed to reduced nicotine concentrations.

Nicotine is a nervous system toxin that can bind to acetylcholine receptors of synaptic cells leading to neuromuscular blockade (*Albuquerque et al., 2009*; *Prochaska and Benowitz, 2016*). Its median lethal dose (LD$_{50}$) varies between 3–50 mg*kg$^{-1}$ for small rodents (*Wink, 1993*; *Karačonji, 2005*). As nicotine is produced at concentrations of up to 3 mg*g FW$^{-1}$ in *N. attenuata* shoots, the consumption of a single wild-type plant can be lethal to a small vertebrate herbivore. Interestingly, certain vertebrates such as the desert woodrat (*Neotoma lepida*) seem to excise *N. attenuata* leaves, dry them and transfer them to their nests (*Baldwin, 2003*). It has been proposed that this behavior may allow woodrats to use nicotine as pharmaceutical means to reduce ectoparasite loads

(*Baldwin, 2003*). Whether nicotine can provide benefits to vertebrates in this way remains to be tested.

In conclusion, our study demonstrates the importance of jasmonates as regulatory signals that boost nicotine production and thereby protect the leaves and stems of *N. attenuata* from a diverse range of herbivores. By studying the interaction between plants and their natural herbivore communities, we show that jasmonates can provide fitness benefits in nature, and that these fitness benefits are derived from protecting vital plant tissues from vertebrates.

## Material and methods

### Establishment of experimental populations

In spring 2012, we established three experimental *N. attenuata* populations ('Lytle', 'Poplar' and 'Snow') across the field station of the Lytle Ranch Preserve (St. George, UT, USA; *Figure 1—figure supplement 1*). Each population consisted of at least twelve jasmonate-deficient inverted repeat allene-oxide cyclase plants (irAOC, line 457) and empty vector controls (EV, line A-03-9-1, 'wild type') that were planted in alternation approximately 1 m apart. The irAOC line has been characterized previously (*Kallenbach et al., 2012*) and was chosen for its strongly reduced jasmonate levels (*Kallenbach et al., 2012*; *Machado et al., 2013*). Plants were germinated and planted as described (*Krügel et al., 2002*; *Schuman et al., 2012*; *Machado et al., 2013*). Seeds of the transformed *N. attenuata* lines were imported under APHIS notification number 07-341-101n and experiments were conducted under notification number 06-242-02r. In the Snow population, half of the plants were induced by wounding the leaves with a pattern wheel and applying *Manduca sexta* oral secretions (W+OS) as described (23). As the induction treatment had no significant effect on herbivore damage or flower production, the parameter was not included in the final statistical models.

### Herbivore damage screen and fitness measurements

Wild type (EV) and jasmonate-deficient irAOC plants from all three populations were screened for herbivore damage five weeks after transplantation. The type of damage was classified as leaf removal, stem peeling, leaf chewing and leaf sucking/piercing and was attributed to different herbivores by using knowledge from previous studies (*Baldwin, 2003*; *Kessler et al., 2004*; *Steppuhn et al., 2004*; *Meldau et al., 2009*; *Stitz et al., 2011*; *Kallenbach et al., 2012*; *Schuman et al., 2012*; *Đinh et al., 2013*; *Schäfer et al., 2013*). The parameters used to determine the presence of herbivores and details about the quantification of the different damage types are summarized in *Figure 1—source datas 1*, *2*. Photographic evidence of some herbivores is provided in *Figure 1—figure supplements 3*, *4*. As a measure of reproductive potential, the numbers of flowers were counted for each plant in the different populations immediately following the herbivore impact assessments (*Van Dam and Baldwin, 2001*; *Glawe et al., 2003*; *Sime and Baldwin, 2003*). Flowers were removed after counting according to APHIS regulations for the use of transgenic plants in the field. In the plot surrounding the Snow population, we observed a spike in gopher activity in the week after the flower count. To be able to include this attack into our statistical model, we reassessed herbivore damage and counted the number of regrowing flowers two weeks later. This second flower count was then used for statistical analysis.

### Fitness costs of induction and defoliation in the glasshouse

To determine the relative value of leaves and stem bark for plant reproduction, we evaluated reproductive output in plants that were subjected to four different regimes of simulated herbivory: simulated insect attack, stem peeling, and full and partial defoliation (*n* =10–12). Wild type (EV) and irAOC plants were grown as described elsewhere (*Krügel et al., 2002*). When plants reached the bolting stage, insect attack by chewing herbivores was mimicked by wounding 3 leaves and applying *M. sexta* oral secretions as described (W+OS) (*Machado et al., 2015*). In another batch of plants, vertebrate herbivory was simulated by removing either the rosette leaves , or by removing both rosette and stem-leaves (i.e, full defoliation). Finally, we mimicked the vertebrate damage patterns observed in the Lytle plot population by removing all leaves and peeling off the bark. To remove the bark, a small incision was made into the epidermis at the stem base. The epidermis and bark were then pulled off with forceps from the bottom to the top, leaving only xylem and pith of the stems

(See below 'Histological staining')(*Figure 2—figure supplement 1*). Intact plants were used as controls. Following the different treatments, all plants were left to grow in a fully randomized setup, and the number of flowers was counted every 3–4 days until the end of the flowering period.

## Plant primary and secondary metabolite profiling

As a first step to understand the mechanism behind the significant differences in vertebrate grazing patterns between wild type (EV) and jasmonate-deficient irAOC plants in the field, we profiled primary and secondary metabolites in the stems and leaves for several plant treatments. Prolonged leaf-wounding and simulated insect attack treatments were carried out as described (*Machado et al., 2013*; *2015*) (*n* = 3–5). Nicotine, chlorogenic acid, rutin and total 17-hydroxygeranyllinalool diterpene glycosides were determined by HPLC-DAD as described (*Keinänen et al., 2001*). Sugars and starch were measured as described (*Velterop and Vos, 2001*; *Smith and Zeeman, 2006*; *Machado et al., 2013*; *2015*).

## Cottontail rabbit feeding preference

To determine which of the jasmonate-dependent changes in host plant chemistry may be responsible for the preference for irAOC plants observed in the field, we developed an *in vitro* feeding assay for cottontail rabbits (*Sylvilagus nuttallii*) as follows.

### Effect of drying process on N. attenuata secondary metabolite profiles

To test the effect of drying on *N. attenuata* secondary metabolite profiles, we measured secondary metabolite levels in rosette stage plants that were both harvested and immediately frozen in liquid nitrogen or were harvested and dried (50°C, 24 hr) prior to secondary metabolite measurements (n = 5). Secondary metabolites analyses were carried out as described (*Ferrieri et al., 2015*). To compare the concentrations between fresh and dried plant materials, we measured water loss during drying and corrected the measured concentrations of the dried materials accordingly.

### Food pellet preparation

Stems and leaves of early elongated EV and irAOC *N. attenuata* plants were harvested and dried (50°C, 24 hr). Dried plant material was then ground using a Thomas Scientific Wiley Mill equipped with a 2-mm sieve (Thomas Scientific, Swedesboro, NJ, USA). The ground material was then mixed (1:5) with Alfalfa Purina rabbit chow (Purina Mills, St. Louis, Missouri, USA) and pelletized by a Buskirk PM605 Lab Mill customized with 3-mm wide holes (Buskirk engineering, Ossian, IN, USA). Alfalfa Purina rabbit chow ('Alfalfa') was used as the rabbit's maintenance food. All three pellet types were divided into two batches. The first batch was left intact and used for the first set of experiments and the second one was used for the nicotine complementation experiments. To complement nicotine levels in irAOC pellets, we first measured nicotine content in the pellets (*Keinänen et al., 2001*) and then, by evenly spraying a solution of nicotine on irAOC pellets, we equalized the nicotine levels of irAOC to those observed in EV pellets (*Figure 4a,b*: insets). EV and alfalfa pellets were sprayed with pure water. Pellets were allowed to dry at room temperature and nicotine levels were measured again (n = 5).

### Feeding preference assay

In a first set of experiments, we tested whether our field observations could be reproduced by our in vitro feeding assay with cottontail rabbits. For this, we presented rabbits with 20 g (±0.1 g) of each pellet type (EV, irAOC and alfalfa) and allowed them to choose and feed freely for a 24 hr period (*Figure 4—figure supplement 1*). Eight rabbits were used in total. One animal was tested at a time in 24 hr-intervals and the full experiment was carried out twice for each rabbit. Food consumption was set as response variable for feeding preference. To quantify food consumption, pellets were placed separately in 9-cm wide and 6-cm deep circular bowls that sat on the top of a balance (A&D EJ3000) (*Figure 4—figure supplement 1*). The balances were adjacent to each other and enclosed in a plywood box (66 cm wide, 38 cm long, and 16 cm high). Automatic recordings of pellet weight were logged on a laptop computer and plotted against time. In a second set of experiments, we tested whether differences in nicotine content between EV and irAOC plants might be responsible for the feeding preferences observed. For this, we carried out a procedure as described above, but

we contrasted feeding preference of cottontails on irAOC+nicotine and EV pellets. The rabbit's maintenance food ('Alfalfa') was used as a control in both sets of experiments. All the choice experiments were carried out using the same rabbits. Trials were approved by the International Care and Use Committees (IACUC) via permit 04495–002. The feeding trials were carried out in the Washington State University's Small Mammal Research Facility, located 4 km east of Pullman, Washington.

## Histological staining

To investigate which tissues were removed from the stems by vertebrates, we simulated this type of herbivory by experimentally removing the epidermal tissue from bolting wild type plants as described above. Peeled and non-peeled stems where then stained with toluene blue for 15 s and photographed using a stereomicroscope equipped with a digital CCD camera (SteREO Discovery. V8, 14 Carl Zeiss Microimaging) and processed with AxioVision LE software (Carl Zeiss 15 Microimaging).

## Statistical analyses

All statistical analyses were performed using the R software (*R Development Core Team, 2015*) unless otherwise stated:

### Leaf-herbivore damage screen and fitness measurements

Since different herbivore communities were observed in the different plots, the effect of damage intensity on flower production was assessed for the three experimental populations separately. ANOVAs were used to compare damage intensity between plant genotypes, for each damage type. The influence of the different damage types on plant fitness (i.e. flower production) was assessed using likelihood ratio tests applied on Generalized Linear Models (GLM; family: negative binomial, link: log) (package 'MASS', function 'glm.nb' (*Venables and Ripley, 2002*). Flower production was compared between plant genotypes using ANOVAs by considering the natural logarithm of the number of flowers as response variable.

### Fitness costs of induction and defoliation in the glasshouse

The effect of the different simulated herbivore treatments on flower production was analyzed separately for each plant genotype using a likelihood ratio test applied on a GLM (family: negative binomial, link: log) considering the number of flowers produced the day of flowering peak as response variable (EV: at day 22 for control, W+OS and rosette defoliation, and at day 26 for full defoliation and stem peeling. irAOC: at day 17 for control, W+OS and rosette defoliation, and at day 26 for full defoliation and stem peeling). Pairwise comparisons of Least Squares Means (LSMeans; package 'lsmeans', function 'lsmeans') (*Lenth, 2015*) were computed using the False Discovery Rate (FDR) (*Benjamini and Hochberg, 1995*) correction for *P*-values. Plant genotypes were compared separately for each treatment using the same procedure, and *P*-values of all tests were FDR-adjusted.

### Plant primary and secondary metabolite profiling

To assess jasmonate-dependent differences in stem and leaf metabolic profiles, a redundancy analysis (RDA), considering as constraints the plant genotype, the treatment and the interaction between these two factors (package 'vegan', function 'rda') was carried out for leaves and stems separately (*Oksanen et al., 2015*). Constraints explained 70.2% and 61.0% of total variance in leaves and stems, respectively. Significance of these three terms was assessed using a permutation test (with 9999 permutations) as described (*Legendre and Legendre, 2012*). The interaction term had a significant effect in both RDAs (p<0.001 in both cases). Metabolites explaining most of the variation between induced EV plants (EV:W+W and EV:W+OS) and the other treatmentgenotype combinations (EV:Control, irAOC:Control, irAOC:W+W and irAOC:W+OS) were identified by computing the Pearson correlation coefficient between chemical data and sample scores by using the directions in the two dimensional vector spaces which discriminate these treatments (axis 1 in the leaves and to the bottom right-to-top left diagonal in the stems). Metabolites with correlation coefficient values > |0.8| were then included as bar graphs in *Figure 3*. To test the effect of simulated herbivory on *N. attenuata* secondary metabolite profiles, two-way ANOVAs with treatment and genotype as factors were carried out with Sigma Plot 12.0 (Systat Software Inc., San Jose,

CA, USA). Levene's and Shapiro–Wilk tests were applied to determine error variance and normality. Holm–Sidak post hoctests were used for multiple comparisons.

## Cottontail rabbit feeding preference

To test rabbit feeding preference, a logistic model was built for each pellet type separately, using the average amount of pellet consumed by 8 rabbits at each time point as response variable. Asymptote values of these models were considered as treatment effects. These asymptotes were compared using their Bonferroni-adjusted 95% confidence intervals. Non-overlapping confidence intervals show significantly different asymptotes. Concentration of nicotine in pellets was compared between treatments using pairwise *t*-tests, using the FDR correction for *P*-values.

## Effect of the drying process on N. attenuata secondary metabolite profiles

To test the effect of drying on *N. attenuata* secondary metabolite profiles, two-way ANOVAs with water status and genotype as factors were carried out with Sigma Plot 12.0 (Systat Software Inc., San Jose, CA, USA). Water-loss corrected values were used for all analyses. Levene's and Shapiro–Wilk tests were applied to determine error variance and normality. Holm–Sidak *post hoc* tests were used for multiple comparisons.

# Acknowledgements

Celia Diezel, Youngjoo Oh and Danny Kessler helped with the field experiments. Abigail Ferrieri and Rajarajan Ramakrishnan supported the metabolite analyses. Arne Weinhold provided photographic evidence for gopher grazing. Pavan Kumar provided photographic evidence for the presence of cottontail rabbits. Lucas Cortés Llorca assisted with the histological staining. The work of ME was supported by a Marie Curie Intra European Fellowship (grant no. 273107). This work was supported by the Max Planck Society and the Collaborative Research Centre ChemBioSys (CRC 1127 ChemBioSys) of the Deutsche Forschungsgemeinschaft (DFG).

# Additional information

### Competing interests

ITB: Senior Editor, *eLife*. The other authors declare that no competing interests exist.

### Funding

| Funder | Grant reference number | Author |
| --- | --- | --- |
| Marie Curie Intra European Scholarship | 273107 | Matthias Erb |
| Deutsche Forschungsgemeinschaft | CRC 1127 ChemBioSys | Ian T Baldwin |
| Institut National de la Recherche Agronomique | | Maxime R Hervé |

The funders had no role in study design, data collection and interpretation, or the decision to submit the work for publication.

### Author contributions

RARM, MM, ME, Conception and design, Acquisition of data, Analysis and interpretation of data, Drafting or revising the article; MRH, Analysis and interpretation of data, Drafting or revising the article; ITB, Conception and design, Analysis and interpretation of data, Drafting or revising the article

### Author ORCIDs

Ian T Baldwin, http://orcid.org/0000-0001-5371-2974
Matthias Erb, http://orcid.org/0000-0002-4446-9834

### Ethics

Animal experimentation: Trials were approved by the International Care and Use Committees (IACUC) via permit 04495-002. The feeding trials were carried out in the Washington State University's Small Mammal Research Facility, located 4 km east of Pullman, Washington. Field experiments: Seeds of the transformed *N. attenuata* lines were imported under APHIS notification number 07-341-101n and experiments were conducted under notification number 06-242-02r.

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
