## [Decision Letter]

Thank you for submitting your article "Benefits of jasmonate-dependent defenses against leaf- and stem-feeding vertebrates in nature" for consideration by *eLife*. Your article has been reviewed by two anonymous peer reviewers, and the evaluation has been overseen by Merijn Kant as Reviewing Editor and Detlef Weigel as the Senior Editor.

The reviewers have discussed the reviews with one another and the Reviewing Editor has drafted this decision to help you prepare a revised submission.

The manuscript by Machado and colleagues makes key advances in understanding how plants modulate their defenses against different herbivores with different feeding styles in nature. It demonstrates how nicotine-based defenses, that are well-known for protecting the plant's flowers and leaves against herbivorous invertebrates, are also established in the plant's stem to protect it against vertebrate herbivores that use a very different feeding-style than their insect counterparts. The value of this work is that it reveals how plants manage to maximize their fitness by customizing a single inducible defense in different organs to combat different attackers. Moreover, this study nicely demonstrates the power of a top-down approach by using a mutant and metabolomics to zoom into the mechanisms which underlie distinct ecological interactions.

While we find the overall story insightful and advancing the field, we do feel that some of the methodology needs to be described into more detail while the importance of the findings and the advance over previous work may not come across optimally.

Firstly, we feel the text does not make clear how the different types of damage were distinguished and how these were quantified. Since there is not a page limit in *eLife* these descriptions could be much more detailed. You describe that you discriminated between damage patterns and that you analyzed these data using analysis of variance but it is unclear what kind of data was put into the analyses. The same accounts for the stem peeling: it is unclear what this is exactly and how it was quantified.

Secondly, there is no information presented on the natural abundances of herbivores during the course of the experiment. Therefore it is unclear to which extent the differences between plots (Results section) can be attributed to absence/presence of different herbivore groups at the different locations rather than absence/presence of jasmonate defenses. If your analyses already take this into account it would be helpful to explicitly mention this in the Discussion. Similarly, we wondered to which extent it is safe to assume that the damage patterns assessed after five to seven weeks are representative also for the period before and after that time. If data to support your assumptions are not available we would like you to indicate in the Discussion why this information is not of importance or to acknowledge more clearly that there may be alternative explanations and/or uncertainties.

Thirdly, we understand why the redundancy analysis was necessary but we do not find the corresponding figure very insightful. A brief explanation of how to read this vector figure would be helpful especially since the figure seems to suggest that if nicotine/DTGs explain the separation between EV and irAOC, then this also holds for rutin/sugars (Figure 3) and glucose/fructose (Figure 3) in the other direction. If so, extra bar graphs for these components will make the output more transparent. Finally, we would prefer an additional (supplemental) table in which the actual chemical data used as input for the RDA analysis are presented as well.

Fourthly, we would like to see a confirmation in the text – e.g. a reference – that counting the number of flowers a couple of weeks after planting is a valid fitness proxy in this system (since these plants will probably flower longer). In addition we were wondering if there is information on (differential) pollinator attraction by the EV and irAOC and, if so, if this may affect fitness independently from herbivory. Any relevant information in relation to this could be mentioned in the Discussion.

Finally, the text as it is now is very compact with very little detail regarding previous studies on secondary metabolites that constrain vertebrate feeding: we feel that the current text may give the reader the impression that deterrence of vertebrates by nicotine is the novelty you are presenting. Therefore we would like to encourage you to discuss previous findings on the effect of secondary metabolites on vertebrates in more detail and to put a bit more emphasis on the actual novelty e.g. that this plant uses a general inducible defense signal to compartmentalize a defense to solve the problem of how to maximize fitness when facing very different kinds of attackers. This will especially be useful to readers searching for the wider context.

---

## [Author Response]

*Firstly, we feel the text does not make clear how the different types of damage were distinguished and how these were quantified. Since there is not a page limit in eLife these descriptions could be much more detailed. You describe that you discriminated between damage patterns and that you analyzed these data using analysis of variance but it is unclear what kind of data was put into the analyses. The same accounts for the stem peeling: it is unclear what this is exactly and how it was quantified.*

We now added a table that provides more information about the different types of damage and how they were quantified ([Supplementary-material SD2-data]). In addition, we provide references to earlier studies on *N. attenuata* which describe the damage types and their associations with different herbivores in more detail. In short, damage types were attributed to different herbivores based on the accumulated knowledge of 15 years of field observations (Dinh et al., 2013; Schäfer et al., 2013; Kallenbach et al., 2012; Schuman et al., 2012; Stitz et al., 2011; Meldau et al., 2009) in combination with direct evidence of herbivore presence (see below).

Secondly, there is no information presented on the natural abundances of herbivores during the course of the experiment. Therefore it is unclear to which extent the differences between plots (Results section) can be attributed to absence/presence of different herbivore groups at the different locations rather than absence/presence of jasmonate defenses. If your analyses already take this into account it would be helpful to explicitly mention this in the Discussion.

To clarify this aspect, we added a table which summarizes how the natural abundance of herbivores (presence/absence) was determined in the three experimental populations ([Supplementary-material SD1-data]). In brief, we characterized the different herbivore communities through a combination of direct sightings, photographs and indirect traces like excrements, eggs and burrows. Characteristic plant damage was also used as an indicator for herbivore presence, but not absence. Note that the observed damage types are strongly correlated with the direct indicators of herbivore presence and that we did not find evidence for herbivore presence without corresponding plant damage in any of the three populations. It is therefore very likely that the differences in jasmonate-dependent benefits between the populations are due to differences in herbivore occurrence.

Similarly, we wondered to which extent it is safe to assume that the damage patterns assessed after five to seven weeks are representative also for the period before and after that time. If data to support your assumptions are not available we would like you to indicate in the Discussion why this information is not of importance or to acknowledge more clearly that there may be alternative explanations and/or uncertainties.

This is a valid point. We assessed damage five to seven weeks after planting (8 weeks post germination) at the flowering stage of *N. attenuata*. As herbivore damage is cumulative, our data covers the entire period from planting to this moment. An earlier study on *N. sylvestris* documents that leaf damage at the elongation stage has the strongest impact on lifetime seed mass (Ohnmeiss and Baldwin, 2000), and experiments with *N. attenuata* show that leaf damage at later developmental stages has little impact on plant fitness (Zavala & Baldwin, 2006). However, we acknowledge that stem borers, flower and seed feeders which occur at later growth stages may still influence the plant’s final reproductive output. This aspect is now discussed in the article.

Thirdly, we understand why the redundancy analysis was necessary but we do not find the corresponding figure very insightful. A brief explanation of how to read this vector figure would be helpful especially since the figure seems to suggest that if nicotine/DTGs explain the separation between EV and irAOC, then this also holds for rutin/sugars (Figure 3) and glucose/fructose (Figure 3) in the other direction. If so, extra bar graphs for these components will make the output more transparent. Finally, we would prefer an additional (supplemental) table in which the actual chemical data used as input for the RDA analysis are presented as well.

We are grateful for these suggestions. We now provide a brief explanation on how to read this figure to the figure legend: “Vectors display the relationship between metabolites and treatments. Vector lengths denote the magnitude of the relationship and the direction whether it is positive or negative”. A supplementary figure (Figure 3—figure supplement 1) shows the means and standard errors of the chemical analyses. The actual chemical data used as input for the RDA analysis will be available through the original datasets that will be deposited with the article.

Furthermore, we have added bar graphs for the metabolites that explain most of the variation between induced EV plants and the other treatment/genotype combinations. To this end, we computed Pearson correlation coefficients between chemical data and sample scores using the direction in the two dimensional vector spaces which discriminates the EV:W+W and EV:W+OS groups from EV:Control, irAOC:Control, irAOC:W+W and irAOC:W+OS groups (axis 1 for the leaves, bottom-right to top left diagonal for the stems; see Figure 3). Metabolites with correlation coefficients > |0.8| were depicted as bar graphs in Figure 3. We now discuss the potential roles of nicotine and soluble sugars in the manuscript.

Fourthly, we would like to see a confirmation in the text – e.g. a reference – that counting the number of flowers a couple of weeks after planting is a valid fitness proxy in this system (since these plants will probably flower longer). In addition we were wondering if there is information on (differential) pollinator attraction by the EV and irAOC and, if so, if this may affect fitness independently from herbivory. Any relevant information in relation to this could be mentioned in the Discussion.

We have added a reference to the text which confirms the strong correlation between the number of flowers and the reproductive output of *N. attenuata* (Hettenhausen et al., 2012; Baldwin et al., 1997). As *N. attenuata* is predominantly self-pollinating (Sime and Baldwin, 2003), we do not expect a strong direct influence of pollinator attraction on plant fitness. We currently have no information on the differential pollinator attraction by the EV and irAOC. Jasmonate-deficient irAOC plants have reduced defenses, increase soluble sugar levels and reduced flower volatile emissions, and all of these factors determine pollinator attraction in *N. attenuata*. For example, nectar nicotine repels hummingbirds (Kessler et al., 2012), while trypsin inhibitors, nectar sugars and benzyl acetone emissions promote natural pollinator visits (Bezzi et al., 2010, Lin et al., 2014). JA-deficient plants will also not produce the morning open flowers that are optimized for hummingbird pollinations (Kessler et al., 2010). Therefore, jasmonate-dependent effects on pollinator attraction are known and expected, but these effects on flower function will be dwarfed by the effects of producing or not producing flowers. We remain cautious in interpreting our flower counts and refer to them only as the reproductive potential.

Finally, the text as it is now is very compact with very little detail regarding previous studies on secondary metabolites that constrain vertebrate feeding: we feel that the current text may give the reader the impression that deterrence of vertebrates by nicotine is the novelty you are presenting. Therefore we would like to encourage you to discuss previous findings on the effect of secondary metabolites on vertebrates in more detail and to put a bit more emphasis on the actual novelty e.g. that this plant uses a general inducible defense signal to compartmentalize a defense to solve the problem of how to maximize fitness when facing very different kinds of attackers. This will especially be useful to readers searching for the wider context.

We are grateful for this suggestion and now mention this aspect in the manuscript. However, we would like to point out that most of the studies on the toxic effects of nicotine to vertebrates has been carried out by direct injections of the metabolite to lab animals (Wink, 1993; Brčić, 2005), and to the best of our knowledge, the results of only one study provide evidence for the potential role of nicotine in protecting plants from vertebrate browsers in nature (Steppuhn et al., 2008). Other secondary metabolites like condensed tannins, phenolics and alkaloids are frequently suggested to determine vertebrate feeding preference in nature (Cooper et al., 1985; Wink 1988; Owen et al., 1993; Furstenburg et al., 1994; O'reilly-Wapstra et al., 2004; Jansen et al., 2007; Fattorusso & Taglialatela-Scafati, 2008; DeGabriel et al., 2009; Rosenthal et al., 2012), but only few manipulative studies are found in the current literature (Jansen et al., 2007; Mkhize et al., 2015), and little is known about whether these metabolites actually improve plant fitness through protection from vertebrates (Foley and Moore, 2005). We therefore believe that the main focus of our study is warranted. We now discuss the literature on the impact of secondary metabolites on vertebrates in more detail.